# Priorities for Research into the Impact of Canine Surgical Sterilisation Programmes for Free-Roaming Dogs: An International Priority Setting Partnership

**DOI:** 10.3390/ani11082250

**Published:** 2021-07-30

**Authors:** Abi Collinson, Marnie L. Brennan, Rachel S. Dean, Jenny Stavisky

**Affiliations:** 1Centre for Evidence-Based Veterinary Medicine, School of Veterinary Medicine and Science, University of Nottingham, Sutton Bonington LE12 5RD, UK; marnie.brennan@nottingham.ac.uk (M.L.B.); jenny.stavisky@nottingham.ac.uk (J.S.); 2VetPartners, York YO30 4UZ, UK; Rachel.Dean@vetpartners.co.uk

**Keywords:** free-roaming dogs, population management, sterilisation, impact, research prioritisation

## Abstract

**Simple Summary:**

Surgical sterilisation is a component of free-roaming dog population management programmes worldwide. However, evidence of the population-level impacts of sterilisation are rarely reported in peer-reviewed literature. Using a priority setting partnership process, we identified the most important unanswered questions concerning these impacts from the perspective of those working with free-roaming dogs. We found that there were many uncertainties surrounding the impacts of such programmes, and how they can be achieved. The top 10 priorities were related to changes in dog population dynamics; risks to human health; human perception and behaviour towards dogs; and logistics related to implementation in the field. Addressing these priorities will enable a more comprehensive understanding of if, how, and why canine surgical sterilisation programmes impact on free-roaming dog populations.

**Abstract:**

Surgical sterilisation is a core activity of free-roaming dog population management (DPM) programmes globally. However, there is limited published evidence on its impact at the population level. To support evidence-based decision making in this field, it is important that research conducted is relevant to those involved in working with free-roaming dogs and implementing such programmes. The aim of this study was to adapt the James Lind Alliance (JLA) user involvement approach to systematically identify the top 10 research priorities regarding the impact of canine sterilisation. International stakeholders with experience working in DPM were asked in an online survey what unanswered questions they had regarding the impact of sterilisation programmes. Thematic analysis of survey responses was used to develop a long list of collated indicative research questions (CIRQs). A literature review was performed to identify questions that were ‘true uncertainties’ (had not been answered by evidence review). These questions were reduced to a shortlist via an online interim prioritisation survey, and a Delphi consensus process determined the top 10 priorities. The top 10 questions related to dog population size and turnover, dog bite incidents, rabies control, implementation in the field and human behaviour change. These priorities were identified and shaped by people with direct experience of canine surgical sterilisation programmes, and as such are an essential resource for directing future funding and research. Addressing these priorities will generate evidence that is directly applicable to policy makers and practitioners who make decisions regarding the management of free-roaming dogs (FRDs) worldwide.

## 1. Introduction

Surgical sterilisation (hereafter referred to as sterilisation) is a common component of humane dog population management (DPM) in countries with free-roaming dogs (FRDs). There are often a number of intended impacts associated with sterilisation, such as controlling population size, improving dog welfare, reducing human–dog conflict and reducing public health risks [1]. Despite the widespread use of sterilisation for several decades in many countries around the world, there is limited published evidence demonstrating if these impacts are achieved [2,3,4]. The use of sterilisation in rabies control is particularly controversial, with recent modelling of different catch-neuter-vaccinate-release (CNVR) intensity scenarios in the field highlighting the potential difficulties in achieving rapid and sufficient dog vaccination coverage to control dog rabies [5]. The use of sterilisation in rabies control is also difficult to evaluate directly, although the potential benefits have been described [4,6]. Therefore, evidence regarding the impact of sterilisation has implications for both animal and human health and well-being. Valid, relevant, and robust evidence is needed by programme managers, policy makers and other stakeholders to make informed decisions regarding the development, implementation and management of sterilisation programmes to ensure they are efficient and effective.

There has been an increased focus across different disciplines on the discrepancy between what users want researchers to study and what researchers actually investigate [7,8]. In human health care, public and patient involvement is widely used in setting research agendas and can be a key facilitator in improving how research activity responds to on the ground requirements [9]. One method of involving users in prioritising unresolved questions about a topic is the James Lind Alliance (JLA) priority setting partnership (PSP) [10]. This methodology was developed to address the mismatch that had been identified between research that was being conducted by academics and the research that the end users of research (e.g., patients and/or clinicians) felt was needed [7,11]. The process can identify novel questions [7,12], and contribute to reduced research waste through increasing the relevance of research produced [13]. The JLA approach involves a stepwise process of establishing a group of representatives from the population of interest to identify and prioritise research uncertainties [10]. The JLA framework has only recently been adapted for use in a veterinary setting, where it has been used to set priorities for research into the treatment of feline chronic kidney disease [14] and equine pituitary pars intermedia dysfunction [15]. However, to date such priority setting has not been applied to a veterinary population management intervention. In the context of DPM, it is possible that the priorities of funding bodies and researchers are not necessarily the same as those of programme managers, veterinarians and other stakeholders who are responsible for planning and implementing sterilisation programmes. Researchers may also be less likely to study practical issues that do not fit into rigorous experimental designs [8], meaning that practitioners may not see the relevance of academic research to their own programme.

Previous research investigating the impacts of sterilisation programmes has demonstrated that many uncertainties remain [2,3,4,6]. This is attributed to a lack of monitoring and evaluation, and the use of designs that do not allow causation to be determined [2,3]. Extrapolation of results and comparison between programmes is hindered by differences in contextual factors (e.g., local dog population dynamics and perceptions of FRDs), implementation of the programme (e.g., capacity, organisational factors), and variations in indicators measured, methods used and study and intervention length [3,4]. Frequently, sterilisation is conducted as one part of a holistic DPM intervention, leading to complications disentangling its specific effects [4,6] from that of impacts mediated by other components, e.g., education initiatives or access to veterinary care. There has also been limited consideration of any unexpected or unwanted impacts of sterilisation, including welfare issues, which may be short term (e.g., surgical complications) or long term (e.g., behavioural or social impacts) [16]. Given the number of potential impacts that could be explored, identification of priority questions for research would be valuable for funders, researchers and DPM implementers to generate relevant evidence that could improve the effectiveness and efficiency of sterilisation programmes.

The aim of this study was to engage individuals with experience working in DPM to identify their top 10 research priorities related to the impact of sterilisation programmes for free-roaming dog populations. This novel approach will provide insights for future research in this field, framed by those who are in a position to use this evidence.

## 2. Materials and Methods

The design, methods, and analysis for this project were adapted from the James Lind Alliance (JLA) priority setting partnership (PSP) process (Figure 1). This is a validated priority setting method that integrates quantitative and qualitative data collection using a stepwise approach [10]. Results were reported using the REporting guideline for PRIority SEtting of health research (REPRISE) guidelines [17] (Appendix A). A protocol (Appendix A) was developed in accordance with the JLA guidelines [10].

### 2.1. The James Lind Alliance Priority Setting Partnership Process Initiation

#### 2.1.1. Steering Group 

The PSP was managed by a research team (AC, JS and MLB) at the Centre for Evidence-based Veterinary Medicine (CEVM), with guidance from a steering group. The process was supported and guided by RSD who has previous experience of conducting adapted PSPs in the field of veterinary science [14,15]. The steering group comprised people with varied experiences of working with FRDs from different geographical and institutional settings, including programme managers of DPM interventions, researchers with experience in FRDs and the Director of an international dog welfare charity.

#### 2.1.2. Participants

The target population was those with current or previous experience of working in the field of DPM. Individuals were eligible for inclusion if they had experience planning, implementing or managing canine sterilisation programmes. A website page was established to advertise the partnership (https://www.nottingham.ac.uk/cevm/population-research/companion-animal-health/impact-of-canine-surgical-sterilisation-programs.aspx (accessed on 27 July 2021)) and provide access to the online survey. Participants were recruited via email and social media using convenience, purposive and snowball sampling in line with the JLA’s inclusive approach. Steering group members promoted the survey through personal and professional contacts. Dogs Trust Worldwide and the Global Alliance for Rabies Control (GARC) also facilitated survey dissemination through their networks.

#### 2.1.3. Scope

The definition of “canine surgical sterilisation programmes” included any initiative in which free-roaming dogs are surgically sterilised. This could involve owners or caregivers presenting dogs, dogs being caught and released, or a combination of both, conducted from a static site (e.g., clinic or shelter), or temporary mobile field sites.

All uncertainties that related to the impact of canine sterilisation programmes on free-roaming dog populations were considered in scope. These could be uncertainties relating to different types of impact, and how to achieve them, or external factors that may affect impacts. Decisions about whether nominated uncertainties were in or out-of-scope were made by the primary researcher (AC) and subsequently verified by the steering group.

### 2.2. Stage 1: Identification of Uncertainties

An online survey was designed using JISC online surveys (Appendix A) (www.onlinesurveys.ac.uk). The survey was available in English. Participants were asked, “What questions do you have about the impact of surgical sterilisation programmes on a population of free-roaming dogs?” If participants were involved in combined sterilisation and rabies vaccination programmes, they were also asked “What questions do you have about using surgical sterilisation as part of a canine rabies control programme?” There was no limit to the number of questions that could be submitted. Participants were also asked to provide some basic demographic information (role, organisation, country of residence and countries in which they work or have worked with FRDs, primary and other aims of their work involving FRDs, e.g., animal welfare, public health).

The survey was pre-tested by members of the Centre for Evidence-based Veterinary Medicine (CEVM) and piloted by the steering group to ensure it was understandable and easy to complete. The text in the survey was refined to provide definitions of key terms used. The final version was circulated via email and social media to organisations and individuals working in DPM by the primary researcher, steering group, Dogs Trust Worldwide and the Global Alliance for Rabies Control (GARC). Participants were also encouraged to share the link with other colleagues or organisations working in this field. The initial survey was open between February and April 2019.

### 2.3. Stage 2: Refining Questions and Literature Review

The aim of this stage was to categorise and refine the submitted questions, verify them as ‘true uncertainties’ (defined by the JLA as questions which have not been answered by systematic review) as opposed to ‘unknown knowns’ (questions which have been answered by published research, but some participants may be unaware of this), and create a list of collated indicative research questions (CIRQs) that were representative of the original submissions.

Responses were downloaded into Microsoft Excel and were anonymised. Each question was screened for relevance by AC based on the inclusion criteria detailed in the protocol (Appendix A). Out of scope questions were excluded. Relevant responses were analysed using a qualitative thematic approach [18] using NVivo (NVivo qualitative data analysis Software; QSR International Pty Ltd., Vancouver, BC, Canada. Version 12, 2018) by AC. This involved initial data immersion (reading and re-reading the submissions), followed by coding of individual responses and grouping of similar codes into themes and subthemes. Collated indicative research questions were created to represent similarly coded responses within each theme/subtheme. For example, “Do sterilised street dogs fight with each other less?” was assigned to a “behaviour/aggression” theme/subtheme and was combined with other similar questions to form the question “Does sterilisation reduce the incidence of dog-to-dog aggression?” The indicative questions were reviewed and verified by the steering group, along with the original submissions, to ensure that they were a true representation, and that the language used would be understandable to all participants.

Each CIRQ was then checked against the existing evidence. A literature search was conducted in CAB Abstracts (1910-present), Medline In-Process and Non-Indexed Citations and Ovid Medline (1946-present) using the OVID interface via the library at the University of Nottingham. Details of the search strategy can be found in Appendix A. The JLA defines a question as considered to be answered if a relevant, up-to-date systematic review or guidelines that address the uncertainty are available. These CIRQs are considered as “unknown knowns” rather than “true uncertainties” and are excluded from the prioritisation process.

### 2.4. Stage 3: Interim Prioritisation

Over 30 uncertainties were identified in the initial survey, and therefore an interim prioritisation survey was necessary in order to have a manageable number in the final process [10]. The aim of this stage was to reduce the long list of CIRQs which were ‘true uncertainties’ (*n* = 47) to a shorter and more manageable list. An online survey was created using card-sorting software (www.optimalworkshop.com). The second online survey was distributed using the same methods as the initial survey and was also emailed to all previous participants who had stated that they were willing to be involved in further stages of this study. The refined list of CIRQs was presented in a randomised order to each participant. Participants were asked to read all of the questions and select (by drag and drop) up to 25 questions that they thought were most important for research (they were not asked to rank the questions). The same demographic information as for the previous survey was also collected. The survey was pre-tested by members of the CEVM and piloted by the steering group prior to launch to ensure that the survey was clear and understandable. Adjustments were made to increase the number of questions that participants could choose and a ‘maybe’ column was added to the ‘prioritise’ and ‘exclude’ options to help with the process of working through the questions. All indicative questions that were moved to the ‘prioritise’ column were ranked based on the frequency with which they were chosen by all participants and by demographic subgroups (e.g., geographical region, organisational setting). The interim prioritisation survey was open from November to December 2019.

### 2.5. Stage 4: Final Prioritisation

In the JLA methodology, the final prioritisation stage is typically conducted using nominal group technique at an in-person one-day workshop. Due to the international diversity, and consequently multiple time zones of participants in this study, it was decided to adapt this stage of the process so that it could be conducted online. The Delphi technique was used involving two rounds of questionnaires to identify the final top 10 priorities. This was conducted electronically; participants were invited to use the online software (www.optimalworkshop.com) via an email link. The data were managed using Microsoft Excel. The rounds were designed to follow the steps used in a traditional JLA priority setting workshop.

The Delphi panel was formed from previous participants who were willing to take part in the final stage of this study. In the first Delphi round (3rd online survey), participants were asked to rank all of the shortlisted questions on a numerical scale with 1 being considered most important. They also had the opportunity to comment on each question outlining why they considered it to be important, or unimportant. The individual rankings were combined to give an aggregate ranking for each question.

In the second Delphi round, each participant was emailed a document with the aggregate ranking and their previous individual ranking for each of the questions. Anonymous comments regarding the reasons why participants considered each question important, or not important were also presented. Participants were asked to re-rank their top 10 questions after consideration of the aggregate ranking and the comments. Suggestions generated in the previous round for rewording of a number of questions were also included and participants were asked to select a preferred version of wording in such cases. Consensus was set at 70% agreement or disagreement for inclusion and exclusion of questions, or for changing original wording to a new version. The individual rankings were combined to form an overall aggregate rank for each question, with the highest scoring questions comprising the top 10. The two rounds were completed between January and April 2020.

## 3. Results

A flow diagram of the process and results are shown in Figure 2.

### 3.1. Participant Demographics

A total of 152 participants from 62 different countries completed one or more stages of this study (see Table 1). Full demographic data can be found in Appendix A. These individuals had been involved in work with free-roaming dogs in 96 different countries. The majority of participants worked for non-governmental organisations (NGOs; 111/152; 73%). A number of participants had multiple roles, e.g., veterinary professional and programme manager of an NGO, or veterinarian working on government-led programme (e.g., as head or coordinators of state rabies control and elimination programmes or other positions within Ministries of Agriculture). The main programmatic aims reported were improving animal welfare, animal health or veterinary public health, followed by human health, education/training, and community engagement. Most participants had multiple aims linked to their sterilisation programme.

### 3.2. Stage 1: Identification of Uncertainties

A total of 110 responses were received from participants in 47 countries, yielding 644 individual questions. The number of questions submitted per participant ranged from 0 to 34, with the median number being 5. Some participants did not submit a question but expressed an opinion or left a comment describing their experiences or asked about their own programme specifically. These were included where relevant statements were made but otherwise were excluded. Questions that were considered too vague, or out of scope were also removed (*n* = 168/644; 26%) (Appendix A), leaving 476 questions remaining. “What is the most humane practical way of catching street dogs?” is an example of a question considered out of scope as it was not related to the impact of the programme.

### 3.3. Stage 2: Refining Questions and Literature Review

Thematic analysis generated nine key themes (Figure 2). These themes were further divided into subthemes (Table 2), and from these 49 CIRQs were developed to represent the original uncertainties submitted. Literature searches conducted on 6th September 2019 (Appendix A) identified two relevant evidence reviews [2,19]. Therefore, two CIRQs were excluded (“Is sterilisation necessary or can vaccination alone be used to control rabies successfully?” and “What are the best indicators to use to measure the impact of a sterilisation programme?”), leaving 47 CIRQs (developed from 458 submissions) taken forward for interim prioritisation (Appendix A). A number of questions (*n* = 30) had more than one part or were relevant to more than one of the CIRQs and therefore the total number of contributing questions is more than the overall number of submitted questions (*n* = 488).

### 3.4. Stage 3: Interim Prioritisation

The interim prioritisation survey was completed by 107 participants (see Table 1 and Appendix A). Of these participants, 41 were new to this study at this stage. After interim prioritisation, the 26 questions (Appendix A) ranked highest overall, and chosen by >50% participants in each demographic subgroup (e.g., geographic region, organisational setting), were taken forward to the final prioritisation stage.

### 3.5. Stage 4: Final Prioritisation

The first round was sent via email to 71 participants from previous stages who had expressed an interest in joining the Delphi panel. A total of 44 participants completed the first round and 36 completed both rounds (see Table 1 for demographic data); 30 of these participants completed all stages of this study. There was one new participant to this study at this stage who was sent the first round by another participant. After the first round, a total of 11 questions were removed as there was >70% consensus that they should not be in the top 10 priorities. Rewording of four of the remaining questions had been suggested and these were displayed to participants in round 2 and individuals were asked to choose the wording they preferred. Free-text comments concerning why individual questions had been ranked as important, or unimportant, were included in the second round. Examples of these comments include “Prioritised as if yes, then great evidence for further support of sterilisation programs from public health funding” (Table 3; Q4) and “Prioritised as we work in a rabies-free country, but rabies is an exotic disease threat. If a rabies incursion were to occur, would be great to know if efforts should focus on sterilisation + vacc, or vacc only, and indeed if our existing sterilisation work is assisting in preparation for incursion event” (Table 3; Q5). After the second round, 8 questions with the highest aggregate scores reached 70% consensus that they should be in the top 10, and a further two questions with the next highest aggregate scores were included to constitute the top 10 research priorities (Table 3). Three questions for which rewording had been suggested were accepted in the amended form, and for the remaining question, the original wording was retained.

There was little change between the two final rounds in the questions making up the top 10, but there were some changes in their order, particularly the middle rankings (Q4–Q7). Only one question that had been in the top 10 after the first round did not make it to the final top 10 (“What factors regulate dog population size, both natural and human mediated?”). The highest ranking question remained in this position throughout all stages of the process, including after the initial scoping survey as it had the most number of submitted questions contributing to its formation. The final priorities fell within five of the nine identified themes: population dynamics (*n* = 3), logistics (*n* = 2), dog behaviour (*n* = 1), rabies control (*n* = 2) and human behaviour change (*n* = 2).

## 4. Discussion

This is the first time that a stakeholder-led framework (JLA) has been applied to identify research priorities for a veterinary population management intervention. The use of this methodology enabled the effective engagement of DPM practitioners globally, and provided a platform for the involvement of end users in directing the research agenda, through the identification and prioritisation of knowledge gaps from their perspective. The ability to reach a widely geographically dispersed participant population using online survey methods, including the modification of the final prioritisation process, was a strength of this study. Therefore, we are confident that the results represent an international viewpoint. The questions submitted suggest that those implementing sterilisation programmes want practical, actionable evidence that will enable informed decision-making in the field and demonstrate their wider value in a One Health [20] context. The research priorities include questions related to the impact of sterilisation on dog population dynamics, risks to public health (dog bites and rabies), and human behaviour change. The process highlighted the existence of gaps in the evidence base regarding both the implementation of programmes and their subsequent impacts. This highlights the importance of measuring relevant impacts which has been previously identified [2,3], but also demonstrates the need for a better understanding of how these impacts can be achieved, and what else may be affecting them. The list of priorities can be used by funders and researchers to prioritise research that answers relevant questions to elicit real-world change.

Despite the widespread use of sterilisation for DPM for several decades, many of the questions submitted highlighted that there is still a need to better understand the pathways through which sterilisation is expected to work for DPM, and how these may interact. Many of the assumptions (such as whether sterilisation alone can result in a smaller, healthier FRD population and a reduction in human–dog conflict) have not been tested. Guidelines exist for indicators of impact and their measurement [21] and this process has confirmed that the indicators included in this document are of importance to stakeholders. Research published to date (see [2] for a comprehensive review) is broadly aligned with the top 10 priority research areas identified by our study. Studies assessing reducing risks to public health, indicators of dog welfare and changes in dog density/population have been most commonly investigated [2]. Of these, changes in direct indicators of improvements in dog welfare such as body condition score and specific health conditions, although on the shortlist of CIRQs, did not feature in our final top 10. This suggests that participants considered changes in population dynamics, and human factors to be more important for improving dog welfare and should be measured. Previous studies assessing changes in human factors in response to an intervention were less common [2]. The importance of human behaviour change in improving dog welfare has been previously identified [22]. Human behaviour change approaches, and the science that guides effective design, implementation and evaluation is emerging, and therefore this lack of evaluation may relate to feasibility, and a lack of necessary expertise to design and conduct good quality studies. For all prioritised impacts, robust, evidence-based conclusions are still lacking and specific questions of relevance to DPM practitioners have not yet been addressed.

Dog population size and turnover, and the most efficient and effective way to achieve changes in these parameters, were considered of highest priority. Questions in this theme accounted for the top 3 questions and were related to 3 of the other top 10 questions (Table 3; Questions (Q) 5, 6 and 7). The top question, which was ranked highest throughout each stage of the process, was “In order to affect the size of a dog population, what proportion of the population needs to be sterilised and over what time period?” Many of the issues associated with FRDs are perceived by stakeholders to be a consequence of overpopulation, so humanely reducing (or stabilising) population size often represents a common goal amongst different stakeholders, regardless of their primary motivations, e.g., animal welfare or public health. A coverage level of at least 70% of the population is sometimes aimed for. However, there is no published evidence for this [1,4]. It is surprising that this key question is, as yet, unanswered. Whilst there is some existing evidence that supports the use of sustained sterilisation to reduce population size in specific areas at rates of 65.7 for females [23] and 61.8–86.5 for males and females [24], there are very few studies in comparison to the number of interventions being conducted. A reduction in population size has also been reported in one community after a year of intervention [25], although sterilisation rate was not reported. The proportion of dogs requiring sterilisation to achieve a reduction in population size is influenced by a number of other factors (which may lie outwith the influence of the intervention), such as human population size and urbanisation [21]. These questions are linked to another CIRQ which was in the final shortlist but did not feature in the top 10 (“What factors regulate dog population size, both natural and human mediated?”). This is a key question, which relates to the top prioritised questions, as it addresses the sources of free-roaming dogs [1] and which factors are maintaining the population. A focus solely on population size may be misleading in terms of the effects that an intervention is having within that population [26], and could potentially underestimate the effects of a programme if used alone. Determinants of population size, and how they interact, is complex. It is therefore recommended that other demographic parameters are also evaluated [26], and roaming dog population density could also be considered [27].

Effects of sterilisation on human perception of and behaviour towards FRDs and public health risks in terms of dog bites and rabies transmission and control accounted for final questions relating to impacts achieved in the top 10. Prioritisation of these questions highlights the intended One Health impacts of these interventions, and recognition of the interconnections between dogs, humans and their shared environment [28]. Interventions that include sterilisation have previously been associated with reductions in dog bite incidence [29,30]. The inclusion of two questions related to rabies control suggests that this is a key goal for those conducting sterilisation programmes in rabies endemic areas, even where main motivations are in improving animal welfare. High population turnover is often cited as the main challenge to maintaining an effective vaccination coverage necessary to achieve a reduction in the incidence of dog rabies [31,32]. This is an example of one of the many links or relationships between questions, and how evidence for one priority (e.g., Q3 “Does a sterilisation programme cause a change in dog population turnover”), would also provide evidence towards answering other uncertainties (Q5 “Do areas with sterilisation and vaccination programmes have a lower incidence of rabies in dogs and humans than areas with vaccination only programmes?”). The questions do not need to be considered in isolation, and indeed may benefit from being considered in combination, or in relation to other questions.

Uncertainties linked to the implementation of programmes (Table 3; Q6 and Q7) reflect examples of operational differences between programmes which are likely to influence their subsequent impacts. An important limitation of the existing evidence base is the degree to which research findings can be directly translated to the diverse implementation strategies employed by practitioners in the field. Implementing sterilisation in the real world is complex, and subject to different constraints (and opportunities) that are unique to any given context. Future research studies must use a suite of methodologies, to account for the complexities of the real world to answer these priority questions. Increased use of interventional (e.g., randomised controlled trials; RCTs) or observational (e.g., cohort studies) study designs to provide evidence of cause and effect has previously been suggested [3]. However, the use of RCTs may not be feasible or ethical for use in the field. The use of before-and-after, interrupted time-series or multiple baseline designs may be more appropriate. However, these study designs may fail to provide sufficient information on how to replicate an intervention in different field contexts.

Evaluation methods that are applicable in real-world contexts and that examine intervention processes are also needed [33]. Process evaluation approaches aim to understand how the intervention was implemented, identify causal mechanisms that impact the outcomes, and the different contextual factors that explain variation in implementation and outcomes [34]. This approach is gathering momentum in human medicine to evaluate complex behaviour change interventions (e.g., cessation of smoking). These evaluation approaches are more holistic, employ a mixed method approach and recognise the complexity of interventions undertaken in the real world. Moreover, the use of realistic evaluation frameworks [35] that aim to identify ‘what works, in which circumstances and for whom?’ [35] rather than simply answering ‘does it work?’ should also be considered. This approach could generate more robust evidence, and therefore improve translation of evaluation findings to other programmes. This identification of the underlying mechanisms that have driven any changes is key to understanding why impacts have occurred and in developing evidence-based strategies for DPM.

### Limitations

The survey was only available online and in English, which may have affected participation. The interim survey had a particularly high number of people access the survey and complete the demographic questions but not the ranking questions (*n* = 64) and it is not possible to say how representative the participants are of the entire population of people working with free-roaming dogs. However, this bias towards participants who use the internet, who could understand written English, and those with a particular interest in the topic under investigation is an accepted limitation of the JLA methodology [10].

A further limitation is that questions may not have been chosen by respondents if they believed that the question had already been answered. This was evident in some of the comments in the free-text section and usually related to unpublished data from one organisation or data from one study. The JLA process requires a more robust evidence base, as a single study can vary in context and quality, and other studies may report different results. In order to mitigate this, it was emphasised in the final round that none of the questions had been answered by research. Interestingly some of the free-text comments had strong opposing opinions for the same question (i.e., some participants felt that particular questions had been answered in a positive way and that the impact did occur, whereas others felt that the impact definitely did not occur). This was most evident in the questions concerning the use of sterilisation in rabies control.

The use of the Delphi process for final prioritisation was a deviation from the traditional JLA methodology. A disadvantage of using this method in place of the standard nominal group technique was the lack of discussion between participants, which can help generate new ideas and fosters a collaborative approach to reaching a consensus [10]. However, the method has been used in another PSP [36] in order to mitigate the risk of ‘loud voices’ being given more importance and resulting in an unrepresentative list of top priorities. Further advantages of using the Delphi process include anonymity of the responses and equal weight given to the opinions of all participants, as well as the ability to give controlled feedback and give participants the opportunity to reassess their initial opinions [37]. The major advantage of this modification for the current study was the ability to complete the process entirely online, and therefore not exclude participants due to their geographical location from being able to participate in the final stage.

This study was the first step in using formal prioritisation methods to identify what is important to stakeholders, and further understand the issues associated with measuring the effectiveness of DPM programmes. Future work should assess the priorities of other stakeholder groups such as funding partners, communities and dog owners/caretakers

## 5. Conclusions

The JLA methodology was successfully adapted to identify the key uncertainties surrounding canine sterilisation interventions from the perspective of those working in the field. Despite their established nature, the implementation of sterilisation programmes and evidence of their impact is hampered by important knowledge gaps. The development of a research agenda to address persisting unknowns is vital to inform and support future interventions to optimise their effectiveness. In addition to impacts of importance, many of the priorities relate to implementation, suggesting that emphasis is needed not only on whether interventions ‘worked’ in achieving certain impacts, but also on how these impacts can be achieved most effectively, and the mechanisms driving any changes. Addressing these priorities, which are applicable to DPM worldwide, will generate relevant evidence to enable a more comprehensive understanding of if, how and why canine sterilisation programmes impact free-roaming dog populations, in the areas of most importance to the end users. This will minimise research waste, improve the translation of research to DPM practice and ultimately benefit both animal health and welfare, and human health and well-being in countries with FRD populations.

## Figures and Tables

**Figure 1 animals-11-02250-f001:**
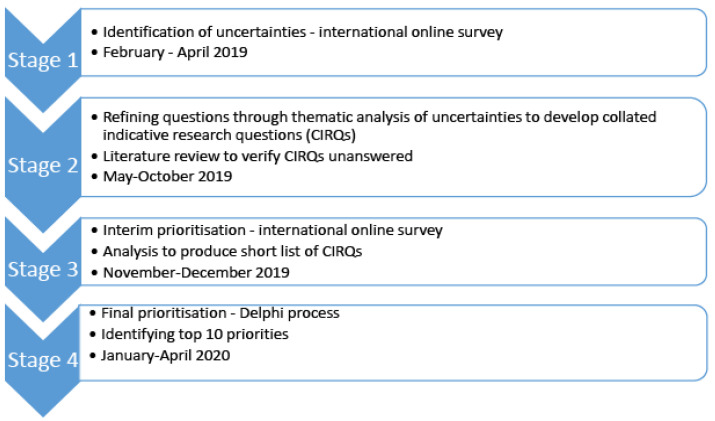
Flow chart of the 4 key stages of the James Lind Alliance methodology as applied to the Canine Surgical Sterilisation Priority Setting Partnership.

**Figure 2 animals-11-02250-f002:**
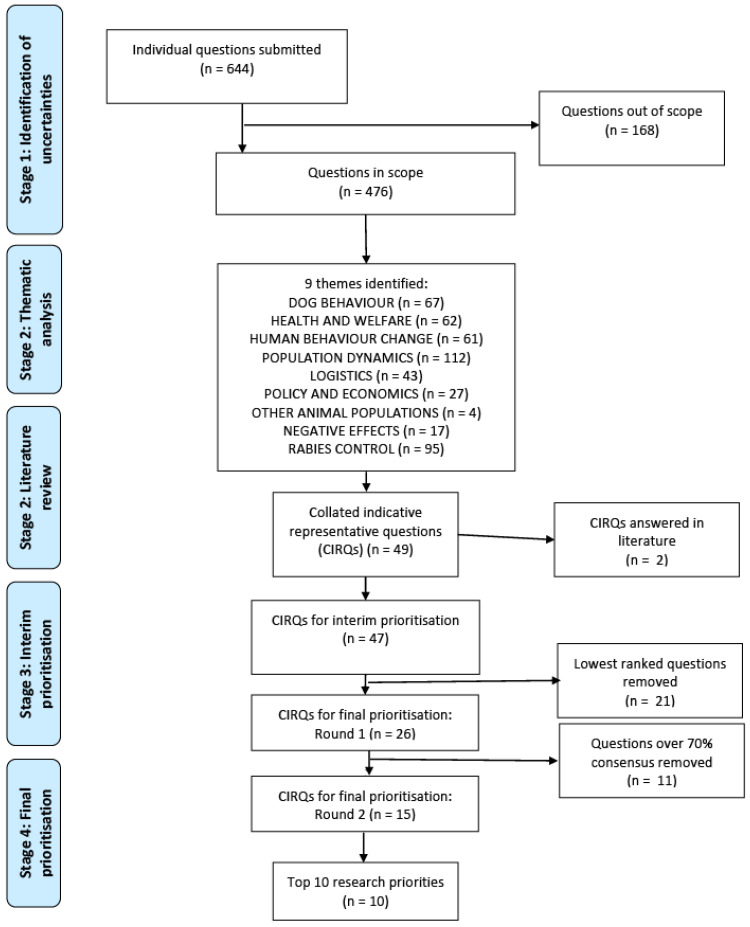
Flow diagram of prioritisation process and evolution of research questions from initial submissions to final prioritisation.

**Table 1 animals-11-02250-t001:** Summary of participant demographics.

		Initial Survey (*n* = 110) %	Interim Prioritisation (*n* = 107) %	Final Prioritisation	
				Round 1 (*n* = 44) %	Round 2 (*n* = 36) %
**Geographical region of residence**	Africa	15	18	11	6
	North America	14	8	11	11
	Latin America and the Caribbean	15	9	16	17
	Asia	28	34	32	39
	Oceania	9	7	11	8
	Europe	20	24	18	19
**Role**	Vet	62	51	39	36
	Vet nurse/technician	10	7	11	11
	Programme manager (non-veterinary)	22	23	32	33
	CAHW ^1^/AHT ^2^	-	5	-	-
	Researcher	3	6	16	17
	Other	4	7	2	3
**Organisation**	NGO	75	71	80	78
	IGO ^3^	3	1	-	-
	Government (national, state)	18	15	5	6
	Academic/research institute	3	12	14	14
	Other	2	1	2	3
**Main aim**	Animal health	26	22	21	19
	Human health/Public health	9	3	2	3
	Public health (veterinary)	18	22	18	17
	Animal welfare	39	44	50	50
	Education/training	2	5	5	6
	Community engagement	2	1	-	-
	Other	4	5	5	6

^1^ Community-based Animal Health Worker. ^2^ Animal Health Technician. ^3^ Inter-Governmental Organisation.

**Table 2 animals-11-02250-t002:** Key themes and subthemes generated from submitted questions.

Key Theme	Subtheme	Example Questions
**Dog behaviour**	Aggression	Do sterilised street dogs bite humans less?
	Roaming	Do sterilised free-roaming dogs roam less?
	Reproductive behaviour	Does a focus on sterilising females have an impact on behaviour of males?
	Social structure/behaviour	Will a sterilised dog be accepted back into its pack after release?
**Dog health and welfare**	Longevity	Average lifespan sterilised free-roaming dogs vs. unsterilised?
	Specific health conditions	What is the impact of sterilisation in preventing TVT ^1^?
	Body condition score	Do sterilised street dogs have a higher body condition score than non-sterilised dogs?
**Human behaviour change**	Barriers and facilitators to participation	How do we better engage with the community stakeholders prior to the project inception to ensure there is full community buy in and involvement?
	Perception of FRDs	Does sterilisation change how people feel about their dogs (increased value)?
	Behaviours towards FRDs	Does this lead to a stronger human-animal bond—improved/more care provided by owners?
**Population dynamics**	Population size	Do sterilisation programs reduce the size of dog populations? What percentage of dogs in a given population of dogs need to be sterilised to keep a dog population stable—no growth?
	Population turnover	Do sterilisation programs reduce the birth rate (puppies born per 1000 dogs in population per year)? Does sterilising a specific dog population prevent other dogs entering this zone?
**Logistics**	Targeting dogs for sterilisation	Who should be the priority for sterilisation and why—young females, adult females, young males, adult males—is this order correct?
	Inaccessible dogs	Is neutering only targeting dogs that are friendly and easy to catch—hence the skittish ones are breeding?
	Geographical considerations	In which dog population group is surgical sterilisation most effective (rural vs. urban)?
	One off/intermittent	Are pop up spay/neuter clinics effective? Do they help to reduce street dog population sizes? How regular do these clinics need to be do be effective?
	Training component	Is there a measurable improvement (or increase) in local veterinary care (either qualifies veterinarians or para-vet personnel)?
**Policy and economics**	Cost-effectiveness	What are the financial impacts for a government to convert from catch and kill to TNR ^2^?
	Changes in policy	Do these programs change government support/intervention attitudes that then enable increased funding to continue such humane programs (vs. mass animal control such as poison baits) long term?
	Stakeholder expectations	Do the realistic expected outcomes of most dog population management interventions match the desired outcomes of stakeholders?
**Other animal populations**	Other domestic animal populations	Do dog population management programs affect the size of the cat population?
	Wildlife	Is sterilisation an effective means to reduce free-ranging dog impact on wildlife?
**Negative effects**	Short term (i.e., directly related to surgery)	What is the surgical complication rate of sterilised free-roaming dogs?
	Long term	Do free-roaming dogs suffer any long-term negative effects of being caught for a sterilisation programme, e.g., greater fear of or aggression towards humans, or difficulty integrating back into their group?
**Rabies control**	Human behaviour change	Would people bring their animals to clinic just for surgery or just for vacc or do they perceive one is better than another?
	Indirect effects	Could sterilisation programs reduce the rate of decline of vaccination coverage in the period between mass rabies vaccination campaigns, and thus allow for an increase in the period between these campaigns?
	Direct effects	Does sterilisation change contact rate between dogs? And therefore reduce rabies risk?
	Logistics	How long does an area that has had a ABC/AR ^3^ intervention of over 70% of the dogs stay rabies free if there is no re-vaccination of the dogs (as per data sheet)?

^1^ Transmissible Venereal Tumour. ^2^ Trap-Neuter-Release. ^3^ Animal Birth Control/Anti-Rabies.

**Table 3 animals-11-02250-t003:** Top 10 research priorities as a result of a James Lind Alliance Priority Setting Partnership and Delphi panel.

Final Ranking	Question	Theme	Aggregate Score
1	In order to affect the size of a dog population, what proportion of the population needs to be sterilised and over what time period?	Population dynamics	279
2	How do sterilisation programmes (of different sizes and durations) affect the size of dog populations?	Population dynamics	221
3	Does a sterilisation programme cause a change in dog population turnover (in terms of birth, death and migration rates)?	Population dynamics	178
4	Do areas with sterilisation programmes have a lower rate of dog bites in people than areas without sterilisation programmes?	Dog behaviour	175
5	Do areas with sterilisation and vaccination programmes have a lower incidence of rabies in dogs and humans than areas with vaccination only programmes (that achieve the same level of vaccination coverage)?	Rabies control	171
6	What are the effects of one-off or intermittent sterilisation (with or without vaccination) programmes in comparison to consistent programmes in an area? What frequency is optimal?	Logistics	163
7	What effect does targeting of female dogs only have on the impact of sterilisation programmes?	Logistics	122
8	Do sterilisation programmes affect community behaviour (human–dog interactions) towards free-roaming dogs?	Human behaviour change	119
9	Do sterilisation programmes affect community perception towards free-roaming dogs?	Human behaviour change	94
10	Does sterilisation have any direct effects on rabies transmission, e.g., in terms of behavioural changes?	Rabies control	87

## Data Availability

All relevant data are within the manuscript and its Appendix A.

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
