# Peer review of "Priorities for Research into the Impact of Canine Surgical Sterilisation Programmes for Free-Roaming Dogs: An International Priority Setting Partnership"

_animals, 2021, doi:10.3390/ani11082250_

Round 1

Reviewer 1 Report

Brief summary: This article aims to use an adapted stepwise, participatory methodology, that has been validated in other research settings, to engage dog population management (DPM) implementers around the world in the research agenda. In doing so, it has successful identified the top 10 priority research questions that dog population management implementers require an answer to, to guide the design, implementation, and evaluation of their work in the field. The authors briefly discuss the difficulties and opportunities surrounding the design and conduct of studies to produce meaningful outcomes. These outcomes must answer important questions and must produce findings that are robust and translatable in the complex, real world contexts where canine sterilisation forms part of a core strategy for DPM intervention.

Broad comments: Overall, this article is successful in applying a priority setting partnership methodology, validated in other research contexts (e.g. human healthcare settings), to identify priorities for research into the efficient and effective implementation of a common veterinary intervention (canine surgical sterilisation). This is a novel methodology for most of the target audience. The authors clearly outline the priority setting process and the stepwise methodology undertaken to engage dog population management stakeholders around the world. The top 10 research questions, as yet unanswered by the published literature, are important to DPM implementers. These research questions can be used to prioritise studies that provide answers to relevant questions to help drive the efficient and effective undertaking of impactful dog sterilisation interventions. A general observation on the manuscript - in places the authors are non-specific when discussing the wider literature in the introduction and the discussion, and the narrative would benefit from in giving examples to support their comments or assertions. My main suggestions relate to clarity and ease of navigation for the readers – many of whom will be new to this methodology.

Specific comments:

Page numbers: The addition of Tables 1 and 3 appear to interrupt consecutive page numbering listed in the header of the manuscript.  

Supplementary Materials: Thank you for supplying the supplementary materials they were useful to understand the steps and complexity of the methodology.

Section entitled ‘Supplementary Materials’, Lines 464-469, and the supplementary material uploaded: I suggest consistent naming of the supplementary file and title given on the first page of the supplementary material and use the same to list the files in the ‘Supplementary Materials’ section.  

Supplementary file name ‘S5 Full demographic data’ has ‘S6 Demographic data of all survey respondents at the top of the page’ – presumably, this should read S5 in the title of the document?

Supplementary file name ‘S6 – Excluded questions submitted to the initial scoping survey’: Tables 1-5 have a column labelled ‘%’ but the column contains no data – I suggest removing the column or reporting %.

Consistent referencing of supplementary materials in the text of the manuscript: Perhaps have a consistent way of referring the reader to the supplementary material in the body of the text e.g., Line 274, I would suggest stating S7 rather than S7 Table.

Figures and Tables:

Figure 1, Line 99: I suggest including full wording in the Figure title rather than the acronym ‘PSP’. The text explaining each stage is small and difficult to read.

Use consistent language relating to stages 1-4 of the Priority Setting Process across figures and manuscript text: I would find it helpful to use consistent language to describe the PSP Stages across

Figures 1 (Line 99) and 2 (Line 232; please also see my comment on Figure 2 below), and the relevant text headers in the method and results sections. For example, Stage 1 is listed as ‘Identifying uncertainties’ in Figure 1 (Line 99), it is then described as ‘Identification of potential research questions’ in the manuscript text (Line 134), and then later described as ‘Identification of uncertainties’ (Line 255).  Similarly for Stage 2, listed as ‘Thematic analysis of uncertainties’, in Figure 1, it is later described as ‘Refining questions and evidence search’ (Lines 152 & 266).

Figure 2, Line 232: I suggest adding another column on the left for consistency to show how each process relates to each of the PSP Stages 1-4.

Figure 2, Line 232: ‘Collated representative questions’ I suggest changing to ‘Collated indicative research question CIRQS for consistency with the text in the methodology and results.

Figure 3, referred to in Line 240:  This Figure is missing in my version of the manuscript. Figure 3 refers to the countries in which participants worked with free-roaming dogs – this information is detailed in Supplementary Material S5, do you need to include a figure with this information? If so, please include, if not, remove the reference to Fig 3 (Line 240).

Table titles: Apologies if this is specific to the journal and cannot be changed. I prefer the title of each table to appear above rather than below the table.

Table 1, Line 252:  Acronyms ‘CAHW/AHT’ and ‘IGO’ are not given in full in the text of the manuscript – I suggest including them in the table footer.

Table 1, Line 252: The data given in the cell: column: ‘Interim prioritisation’ x row: ‘Role’, totals more than 100% - are the roles mutually exclusive? Or can 1 participant’s data be represented in more than 1 role? Please check data or make a note in the table legend if the categories are not mutually exclusive.

Table 2, Line 276:  Theme no 4 entitled ‘Population size and structure’, the corresponding subtheme listed is ‘population size’ and ‘population turnover’. Turnover and structure are not the same, and they are often used interchangeably in the text of the manuscript. I suggest the term ‘dynamics’ is more accurate as a descriptor for the main theme rather than ‘structure’. I suggest using ‘turnover’ rather than ‘structure’ in the body of the text and in Table 3, column 3, Line 306.

Table 2, Line 276: I suggest the headers of each column are consistent with the text in Lines 266-276, ‘Theme’ suggest ‘Key Theme’.  

Main manuscript text:

Simple summary: Line 10, I suggest changing the term ‘common part’ to ‘component’ or ‘critical component’.

Line 14: Please state who was represented from the field.

Line 15: Suggest the addition of ‘identified’ before ‘top 10 priorities’ to emphasize their identification by participants.

Line 16: Suggest replacing ‘attitudes’ for ‘perception’ to be consistent with the language used in the questions identified by participants (e.g., Table 3, Question 9, Line 306). Suggest adding ‘towards dogs’ after human behaviour.

Lines 15-16: Suggest consistent ordering of Top 10 priorities with those ranked in Table 3 (e.g., ‘change in population size and turnover; risks to human health; human behaviour; logistics related to implementation in the field’.)

‘Attitudes’ and ‘perceptions’ are used interchangeably throughout the text. I would suggest only referring to ‘perceptions’ to use consistent terminology with participants returned questions. Although attitudes and perceptions are interrelated, they appear to have distinct definitions in the psychological literature – I am not sure it is accurate to use them interchangeably?

Abstract: Lines 21-22: I suggest rewording and shortening this sentence to ‘However, there is limited published evidence on its impact at population level’, to reflect only published evidence has been reviewed (many orgs may have a wealth of data that is yet to be published in the peer reviewed literature!) Further, surgical sterilisation is effective at controlling the reproduction of individuals, what is lacking is published evidence regarding impact at population level to control rabies and to reduce the size of dog populations (often stated as desired impacts by stakeholders).

Lines 24-25: I suggest emphasizing that the method is a four-stage process, used to systematically identify the top 10 research priorities regarding the impact of canine sterilisation by experienced DPM implementers.

Lines 25-31: I suggest finding a more elegant way to better emphasize the order of the different stages – for example, an online survey was conducted first, thematic analysis followed, an evidence search was performed next etc.

Line 29:  The term ‘true uncertainties’ is not clear (it is once you’ve read the methodology). I would suggest ‘An evidence search was performed to identify whether the questions were true uncertainties (had not been answered by systematic review), rather than ‘unknown knowns’ (due to participants lack of awareness of published research) – these questions were subsequently removed from further analysis’. Next sentence to outline the interim prioritisation survey.

Line 30: States…’and the remaining questions were ranked in an interim prioritisation survey’, yet Lines 190-191 of the Material and Methods section, subheading ‘Stage 3: Interim prioritisation’, specifically state ‘(they were not asked to rank the questions).

Lines 31-32: Include ‘implementation in the field’ between ‘rabies control’ and ‘human behaviour’ to reflect the final questions listed in Table 3 (Line 306).

Keywords: I suggest including ‘impact’ and ‘research prioritisation’ and removing ‘animal birth control’ to give a more accurate reflection of the topics included in the manuscript.

Introduction:

Line 40: I suggest including ‘Surgical sterilisation (hereafter referred to sterilisation)’, then removing ‘surgical’ in the rest of the text.

Line 42: ‘Intended aims’ – the term ‘aims’ and ‘impact’ are used interchangeably throughout the document. For consistency I suggest using the term ‘impact’ rather than ‘aims’.

Line 43: Reference [1] – ICAM have updated this guidance. Include the updated reference: ICAM Humane Dog Population Management Guidance, Updated 2019 (access here).

Lines 46-48: It would be helpful to be more specific about the controversies and benefits of sterilisation in rabies control e.g., the use of sterilisation in rabies control is difficult to evaluate directly, and recent modelling of different intensity CNVR scenarios in the field highlight the potential difficulties in achieving rapid and sufficient dog vaccination coverage to control dog rabies.

Lines 49-52: I would suggest minor rewording to emphases the quality of evidence required to make decisions, who the decision makers are, and how evidence is used e.g.,  ‘Valid, relevant, and robust evidence is needed by programme managers, policy makers and other stakeholders to make informed decisions regarding the development, implementation and management programmes to ensure they are efficient and effective’. 

Line 55-57: Include a citation for sentence that outlines the importance of the involvement of public and patients in setting research agendas etc.

Lines 59-61: For clarity, I suggest rewording this sentence e.g., ‘This methodology was developed to address the mismatch between research that was being conducted (by academics?) and the research that end-users (e.g. patients requiring treatment) felt was needed.'

I note there is an important distinction between the human patient end-users and the participants included for this research. For example, the main beneficiaries of DPM interventions are the dogs themselves and the people who share their lives (e.g. the community, care-takers or owners), yet their views cannot be (e.g. dogs) or are not directly represented (e.g. the people who share their lives) by the study…the conservation example [8] may be more relevant?

Line 75-76: Please include references to support the sentence ‘Previous research investigating the impacts has demonstrated that many uncertainties remain.’

Lines 82-83: Rather than ‘mixed intervention’ I suggest ‘as part of a more holistic DPM intervention (e.g., ICAM 2019), leading to complications disentangling its specific effects [4,6] from that of other impacts mediated by …list other components that may confound or enhance sterilisation efforts?

Line 84: Welfare issues, I would suggest being more specific. [15] refers to a suite of potential detrimental effects of poorly conducted sterilisation for dog welfare.

Line 85-86: Who would find these priority questions for research valuable and what would they use them for?

Materials and Methods: The materials and methods section clearly outlines the stages of the methodology. I suggest minor revisions to the text to aid clarity for the reader.

Line 103: Do you need to mention ethical review here? It is clearly stated elsewhere e.g., Institutional Review Board Statement (Lines 476-479). If needed in the Materials and Methods section, I suggest moving it to Line 93. ER is an inherent requirement of all studies, not specific to ‘Establishing PSP’.

Line 93: Can you include a subheading for clarity? E.g., ‘The James Land Alliance Priority Setting Process’.

 Line 95: I suggest including ‘using a stepwise approach’ after ‘types of data collection’.

Lines 111-112: I suggest moving the following sentence ‘A protocol was developed in accordance with the JLA guidelines [9] (S2) and adapted for use in this novel setting.’ to Line 101 before the sub-heading ‘Establishing PSP’. The protocol given in S2 outlines the establishment of PSP.

Line 102: Give ‘PSP’ in full in the subheading, rather than an acronym.

Line 103: I suggest, if possible, adding another subheading below ‘Establishing PSP’, entitled ‘The Steering Committee’ as this seems to be an important and distinct step in the PSP.

Lines 105-109: I suggest rewording the sentence e.g. ‘The steering group comprised people with varied experiences of working with FRDs from different geographical and institutional settings, including programme managers of DPM interventions, researchers with experience in FRDs and the Director of international dog welfare charity. Delete the next sentence (Lines108-109).

Lines 116-117: I suggest including, ‘and provide access to’ the online survey. If possible also include a link to the website.

Lines 128-129: Add ‘Initially’ at the start of the sentence e.g. ‘Initially all uncertainties..’, to clarify that all uncertainties provided by participants were considered, before the primary researcher and steering committed screened the responses to verify whether they were in or out of scope. How was impact defined? In the participants questionnaire (S3) the term ‘aim’ appears to replace the term ‘impact’.

Lines 129-131: I suggest simplifying the sentence for clarity e.g. ‘ These could be uncertainties relating to different types of impact, and how to achieve them.’

Lines 131-133: I suggest ‘Further decisions about whether..’ to emphasize that another revision step took place.

Line 135: Include link to website for JISC online surveys. I suggest adding a sentence ‘The survey was available in English’.

Line 191: There are 26 questions listed in S8, but the text states ‘..up to 25 questions’. Why the discrepancy?

Line 206: I suggest removing ‘in order’.

Line 206-207: I suggest the following revision for clarity: ‘This was conducted electronically; participants were invited to use the online software (www.optimalworkshop.com) via an email link’.

Line 208: Was the process or the data managed using Microsoft Excel?

Lines 208-209: I suggest removing the following sentence, ‘The rounds were designed to follow…’ , you have explained the necessity for the deviation in the JLA methodology.

Line 211-212: I suggest simplifying the sentence: ‘The Delphi panel was formed from previous participants who were willing to take part in the final stage of the study’.

Line 215: Remove ‘and’ and replace with ‘outlining’.

Lines 218-228: This is an important part of the method, but I found this paragraph very difficult to follow. Can this paragraph be simplified using shorter sentences?

Results: The results section was clearly presented. I suggest minor revisions to the text to aid clarity for the reader.

Line 237: I counted 62 different countries listed in Supplementary Material ‘S5’.

Lines 244-247: These two sentences could be simplified and combined? e.g. ‘The main programmatic aims reported were improving animal welfare, animal health or veterinary public health, followed by human health, education/training, and community engagement.’ I suggest removing ‘wildlife conservation’ – it is not listed separately in Table 1.

Line 249: Demographic data of survey respondents appears in Supplementary Material, file name ‘S5 Full demographic data’ not supplementary file 4 as written in the text.

Lines 256-257: For clarity, I suggest: removing ‘and’ (Line 256) replace with ‘yielding’; remove ‘were submitted’ replace with ‘for stage 1: identifying uncertainties’.

Line 270: Please give citations for the two reviews identified by the literature search.

Line 282: I suggest referring the reader to Supplementary Material ‘S8’ after ’26 questions’ rather than at the end of the sentence (Line 284).

Discussion:

Line 319: Suggest adding ‘in the field’ after ‘…informed decision-making’.

Line 321: Capitalisation: One Health. The latter part of the sentence referring to One Health, seems out of place with the first half of the sentence relating to what participants wanted.

Line 324: I suggest removing ‘as well as additional elements that may affect these impacts.’ – I’m not sure which additional elements you are referring to – are they in addition to the research priorities listed in Lines 321-322?

Lines 327-329: This sentence feels a bit clumsy - questions do not have the capacity to guide funders. The list of priorities can be used by scientists and funders to prioritise research that answers important (relevant) questions to elicit real-world change.  

Lines 330-334: I suggest rewriting this sentence. In addition: remove the term ‘humane’ (Line 330). In the field, the practice of surgical sterilisation may be far from humane. The term ‘core’ (Line 332) before dog population demographics is unnecessary. What is the general theory of change? Please describe and provide a reference.

Lines 336-339: I suggest rewriting this sentence to improve clarity. For example, ‘Research published to date (see [2] for a comprehensive review) is broadly aligned with the top 10 priority research areas identified by our study. Studies assessing the effectiveness of interventions aimed at reducing risks to public health and dog density/population appear to be prioritised over other impacts. Indeed, these types of studies are well designed and conducted’ – state the consequences of ‘higher quality designs’ e.g., do they produce reliable results, do they help to identify causation?

Lines 339-341: Start the sentence with ‘Conversely’, to link it to the previous sentences. I note that human behaviour change approaches, and the science that guides effective design, implementation and evaluation is emerging and gathering momentum. This apparent lack of importance may therefore relate to feasibility and a lack of necessary expertise to design and conduct good quality studies.

Line 341-345: I suggest rewriting the first part of the sentence, it’s repetitive of the previous sentence. Please cite the previous research you refer to. I don’t think you need to specifically state BCS and specific health conditions are welfare indicators – this is implied in the first part of the sentence.

Lines 345-348: I suggest shortening the last part of the sentence,  e.g., ‘… are considered important for improving dog welfare and should be measured.’

Line 348: I am not sure what you mean by ‘areas’.

Line 350: To be specific, replace ‘these’ with ‘DPM’.

Line 351: Remove ‘the most timely and pressing’. The term prioritisation and prioritisation process imply they are important.

Line 356: Que 4 rather than Que 5 listed in Table 3 appears to be more relevant.

Line 359: I suggest removing ‘thought’ and adding ‘perceived by stakeholders to be a consequence of overpopulation…’. Overpopulation is often perceived rather than based any data and objective threshold.

Line 360: Replace ‘common ground’ with ‘common impact or aim or goal’.

Line 362: Who suggests sterilising 70% of the population?

Line 363: Include ‘published’ before ‘evidence’ – there may be unpublished research out there, from which the 70% threshold has been derived.

Lines 364-366: Line 364: Include ‘sustained’ before ‘use of sterilisation’, both studies cited were based on continuous intervention spanning years. Can you include the sterilisation rate that was reported in these studies? Line 365: The authors [18,19] don’t appear to specifically mention density in their publications; only changes in population size are reported.

Lines 366-368: Please be more specific. I suggest ‘The proportion of dogs that should be sterilised to achieve a reduction in the dog population size is influenced by a number of other factors (which may lie out with the influence of the intervention), such as human population size and urbanisation (ICAM 2015) and the availability of owned roaming dogs for sterilisation.’

Line 372: I suggest including a citation for ‘source of free roaming dogs’ (see ICAM 2019, p17). This is clearly illustrated in the ICAM publication.

Line 372: Please include a citation or citations for the previous research that you refer to.

Lines 375-377: Do you have any specific examples (please reference), that demonstrate the potential changes you describe that may be observed with reduced birth rates?

Lines 384-386: Perhaps include Macpherson et al 2013 Dogs, Zoonoses and Public Health 2nd Edition as an example of the interconnectedness (and the consequences of shared environments)?

Lines 388-389: Include ‘an effective’ before ‘vaccination coverage’ followed by ‘necessary to achieve a reduction in the incidence of dog rabies’.

Lines 399-402: Could these sentences be rephrased? I think what you’re trying to say is: An important limitation of the existing evidence base is their relevance, and the degree to which research findings can be directly translated to the diverse implementation strategies employed by practitioners in the field. Implementing dog sterilisation in the real world is complex, and subject to different constraints (and opportunities) that are unique to any given context; thus a one-size-fits-all answer to priority questions is not appropriate or valid.

Lines 402-404: Could this sentence be rephrased? I think the key points are: Research outcomes need to be meaningful to answer priority questions that individuals/organisations can use to design and implement impactful dog sterilisation programmes. Future research studies must use a suite of methodologies, to account for the complexities of the real world to answer these important questions.

Line 405: I suggest, ‘To provide evidence of cause and effect’ rather than ‘to better determine’ is more specific. The use of RCTs may not be feasible or ethical for use in the field. These study designs may also fail to provide sufficient information on how to replicate an intervention in different field contexts.

Lines 407-410: The use of process evaluation approaches deserves a clearer description in the text. The key points are – the approach is gathering momentum in human medicine, to evaluate complex behaviour change interventions (e.g. cessation of smoking). These evaluation approaches are more holistic and recognise the complexity of interventions undertaken in the real world; they aim to understand how the intervention was implemented, identify causal mechanisms that impact outcomes and the different contextual factors that explain variation in implementation and outcomes. They employ a mixed methods approach deriving evidence from quantitative and qualitative data.

Lines 410-413: These two sentence could be more concise e.g., ‘Moreover, the use of realistic evaluation frameworks, that aim to identify what works, in which circumstances and for whom?, rather than simply ..'

Lines 413-415: Missing a couple of critical points? The approach could generate more robust evidence, and therefore improve translation of evaluation findings to other programmes.

Line 424: Add, ’who could understand written English’, after ‘who use the internet’.

Lines 429-431: Replace ‘a stronger evidence base’ with ‘a more robust evidence base’.

Line 442: I suggest starting the sentence with ‘Further advantages of using the Delphi process include…’

Reviewer 2 Report

I don’t remember the last time I reviewed a manuscript as well written, clear, thoughtful and potentially impactful as this one.

My only suggestion is to try and make the font larger in figure 1.

Brava!

Reviewer 3 Report

Please see report attached

Reviewer 4 Report

This paper considers canine surgical sterilisation within free-roaming dog populations and aims to identify the top 10 unanswered questions, thus priorities for research, within this topic. To achieve this, the authors adapted the James Lind Alliance (JLA) method for identifying research priorities. This involved collecting qualitative data through online surveys of international stakeholders to identify uncertainties then using the Delphi consensus process to identify the top 10 priorities. The paper concludes that there are many uncertainties concerning impacts of surgical sterilisation programmes and that these relate to changes in dog population size, human attitudes and behaviour, and risks to human health. These are clearly important findings and highlight the need to address these priorities, with a focus on importance to the end users. Addressing these priorities should allow stakeholders to better understand the impacts of surgical sterilisation on, and management of, free-roaming dog populations.

This paper is well written, concise, and easy to understand. The introduction provides a solid overview to the topic and is aided by sufficient referencing. It clearly outlines the issues concerning free-roaming dogs and human health and the need for further research to support evidence-based decisions and policies. The authors make a number of important and thought provoking points, including the discrepancy between the research that stakeholders want and the research that is actually undertaken. Hence, the authors explain why they chose the JLA method and also briefly detail where this method has previously been used but acknowledge that, interestingly, this is the first time such priority setting to veterinary intervention (is there any known reason for this?)

The authors clearly describe the methods used. This is aided by a flow chart. The supplementary materials provide useful information about methods used (e.g. guidelines), the initial survey, and the development of questions used (or not used) at different stages – it’s good to see these included as it is so important to understand more about these and show how this work progressed. There are aspects of the methods where I would like to see more detail, e.g. whether an individual was responsible for all thematic analysis or whether this was undertaken by a team. These are noted in my specific comments.

In the first online survey, the two main survey questions (S3; section 2 and section 3) both give examples of questions that participants might have. Have the authors considered any bias, e.g. whether the inclusion of these example questions encouraged respondents to suggest similar questions themselves?

The results, discussion, and conclusions clearly highlight the main findings and importance of this work. The authors acknowledge a number of limitations but these do not prevent this research from providing a very useful guide to prioritisation areas for future work.

I have provided a small number of specific comments below. Most of these are very minor errors and a small number are my personal preference hence are suggestions and may not require any changes.

Line 99: Figure 1. The writing is not very clear in the flow chart/it looks like a low quality image? (Apologies, this might be a formatting issue that will be rectified before publication.)

Line 135: Can you give more details about the where this survey was hosted? E.g. was this through an external survey website or hosted by the University of Nottingham, etc. (These details are provided for the second online survey on line 185.)

Line 160: Who analysed the initial survey data? (And, although not noted again, any other survey data/free text responses, e.g. comments in surveys 3 & 4.) On Line 132 you note that the primary researcher made decisions about inclusion in scope. Similar recognition would be useful throughout the methods. Additionally, it is important to understand whether all coding was undertaken by one individual or whether this was shared – and discussed – across a team, e.g. separate coders analysing different text and combining codes plus using discussions between coders to discuss the meaning of text and develop themes/subthemes, etc. and /or if the same text was coded independently by different researchers, e.g. to test reliability by comparing the coding of two independent researchers or “sense checking” coding with another researcher. In lines 169-171 you note that the questions were reviewed by the steering group (great!) but this is also important at the initial qualitative analysis stage.

Line 161: Any reference(s) for qualitative thematic analysis? Similarly, although a (good) brief overview is provided on lines 162-164, referencing would be helpful.

Line 180: Interim prioritisation section. It might be useful to include something about this second survey as supplementary materials. Although I appreciate this survey is described in this section (and the questions are later referred to as being in S7) I would like to see a visualisation of what the survey looked like and have an idea of the questions (or at least reference to these being in the SM). This would also be useful for understanding of the following section (final prioritisation).

Line 249: Is the naming of the supplementary files correct? Text states “supplementary file 4 (S5)”. The actual document is named S5 but the title in it is S6?! (Also, within the S5 file, there is inconsistent use of bold for column headers in tables. Additionally “Table” appears to refer to different stages as opposed to actual tables. It might be useful to change this, should readers wish to refer to specific tables.)

Line 277: Table 2 clearly sets out themes and subthemes. I feel that a third column including quotes to demonstrate these themes/subthemes would enhance this paper (these quotes really are your raw data and at the core of this paper [plus – as someone who does quite a bit of qual research – I do enjoy reading quotes!]). If not in the main paper, perhaps quotes related to these themes/subthemes could be included as SM?

Line 317: Although I agree that achieving an international viewpoint is impressive and important, I’m curious as to whether you detected (or were able to look for) and differences between countries? Are differences between cultures/countries important to acknowledge?

Line 320: Phrase “one health” not capitalised here but it is capitalised in Line 384. Possibly a reference would be useful for context.

Lines 330-333: This is quite a long sentence. I’m not sure that the first comma on Line 332 is needed. (I think “both” refers to 1) fundamental process and 2) core dog population demographics but I had to read the sentence a few times. Possibly consider rewording/splitting into two sentences for readability.)

Line 422: Do you know there were so many who completed the demographic questions but not the ranking?

Lines 426-434. This is a really important point. I’d be interesting to know more (generally) about the content of the free text responses and how they were analysed, e.g. where these coded, did any themes emerge, etc.? Is this in any of the supplementary files or could this be added?

Overall a really interesting and well written paper using a novel approach to address a real world issue. I look forward to seeing this published and hope it will lead to further research into this important area.
